# On the Transition from Control Modes to Spontaneous Modes during ECMO

**DOI:** 10.3390/jcm10051001

**Published:** 2021-03-02

**Authors:** Krista Stephens, Nathan Mitchell, Sean Overton, Joseph E. Tonna

**Affiliations:** 1Department of Emergency Medicine, University of New Mexico, Albuquerque, NM 87131, USA; KrStephens@salud.unm.edu; 2Division of Emergency Medicine, Department of Surgery, University of Utah Health, Salt Lake City, UT 84132, USA; u0837115@utah.edu; 3Division of Critical Care, Department of Anesthesiology, University of Utah Health, Salt Lake City, UT 84132, USA; sean.overton@hsc.utah.edu; 4Division of Cardiothoracic Surgery, Department of Surgery, University of Utah Health, Salt Lake City, UT 84132, USA

**Keywords:** mechanical ventilation, spontaneous breathing, ARDS, respiratory failure, ECMO

## Abstract

The transition from control modes to spontaneous modes is ubiquitous for mechanically ventilated patients yet there is little data describing the changes and patterns that occur to breathing during this transition for patients on ECMO. We identified high fidelity data among a diverse cohort of 419 mechanically ventilated patients on ECMO. We examined every ventilator change, describing the differences in >30,000 sets of original ventilator observations, focused around the time of transition from control modes to spontaneous modes. We performed multivariate regression with mixed effects, clustered by patient, to examine changes in ventilator characteristics within patients, including a subset among patients with low compliance (<30 milliliters (mL)/centimeters water (cmH_2_O)). We found that during the transition to spontaneous modes among patients with low compliance, patients exhibited greater tidal volumes (471 mL (364,585) vs. 425 mL (320,527); *p* < 0.0001), higher respiratory rate (23 breaths per minute (bpm) (18,28) vs. 18 bpm (14,23); *p* = 0.003), greater mechanical power (elastic component) (0.08 mL/(cmH_2_O × minute) (0.05,0.12) vs. 0.05 mL/(cmH_2_O × minute) (0.02,0.09); *p* < 0.0001) (range 0 to 1.4), and lower positive end expiratory pressure (PEEP) (6 cmH_2_O (5,8) vs. 10 cmH_2_O (8,11); *p* < 0.0001). For patients on control modes, the combination of increased tidal volume and increased respiratory rate was temporally associated with significantly low partial pressure of arterial oxygen (PaO_2_)/fraction of inspired oxygen (FiO_2_) ratio (*p* < 0.0001). These changes in ventilator parameters warrant prospective study, as they may be associated with worsened lung injury.

## 1. Introduction

The transition from control modes to spontaneous modes is ubiquitous during critical care, occurring among almost all mechanically ventilated patients. Spontaneous modes always differ from control modes in at least the absence of all of the following: (1) a fixed inspiratory duration, (2) a minimum respiratory rate, and (3) for volume regulated modes, a fixed tidal volume. For patients with improving respiratory function and stable metabolic demands, these differences may not be relevant and observed ventilatory parameters on spontaneous modes may remain unchanged. In contrast, for patients with lung injury, the loss of a fixed inspiratory duration and allowance of an unregulated tidal volume may lead to tachypnea, increased patient effort, and changes in tidal volume.

While there are studies to inform optimal liberation from mechanical ventilation [1,2], there are no studies we identified among patients managed on extracorporeal membrane oxygenation (ECMO). The significance of the concomitant use of ECMO during mechanical ventilation cannot be overstated, and relates primarily to the fact that patients on ECMO can have gas exchange independent of the ventilator. On ECMO, two of the primary drivers of respiratory effort—partial pressure of arterial carbon dioxide (PaCO_2_) and partial pressure of arterial oxygen (PaO_2_)—can be modified independently from the ventilator circuit. ECMO thus enables a greater range of ventilator settings—such as respiratory rate and tidal volume—than would be possible without ECMO. As both respiratory rate and tidal volume are important to the development of lung injury [3,4,5,6,7,8,9], we wished to understand ventilator parameters while on ECMO, specifically during the transition from control modes to spontaneous modes.

Our goals were to determine, using high fidelity data among a large cohort of patients on ECMO, (1) if there were differences between the ventilatory parameters during spontaneous modes compared to control modes, (2) if there was a pattern to these differences, (3) if these differences were associated with meaningful clinical changes at the level of the lung, such as changes in oxygenation, excessive tidal volume, or increases in respiratory rate.

## 2. Experimental Section

### 2.1. Data Source and Study Population

This secondary analysis was approved by the Institutional Review Board at the University of Utah under #00101562. Patients who received mechanical circulatory support or extracorporeal membrane oxygenation (ECMO) at the University of Utah from 1st January 2010 until 11th April 2019 were identified from the institutional mechanical circulatory support registry and included patients with both venoveno and venoarterial ECMO. Ventilatory data from these patients were then extracted from the University of Utah Electronic Data Warehouse (EDW) by a data scientist blinded to the goals of the analysis. The University of Utah EDW includes all electronic medical record (EMR) entries, including respiratory therapy and ventilator data; it has been previously validated as sufficiently accurate and complete for research [10,11]. Ventilator data is entered into the EMR by a clinical respiratory therapist, who record all changes to the ventilator, including date and time of change and who are in house 24/7. Patients were included in the analysis if they were ≥18 years of age and received ECMO and mechanical ventilation during their admission. We excluded patients who were not mechanically ventilated, who received only non-invasive positive pressure ventilation, who were missing at least one variable for analysis, and we excluded repeat ECMO runs. Appendix A details the patient flowchart.

### 2.2. Clinical Variables

Variables for analysis included both recorded and calculated variables. Recorded variables included mode (volume control, pressure control, spontaneous, and T-piece (for patients with a tracheostomy), respiratory rate in breaths per minute (BPM), tidal volume (V_T_) in milliliters (mL), PaO_2_, fraction of inspired oxygen (FiO_2_) (from 0.21 to 1), positive end expiratory pressure (PEEP) in centimeters of water (cmH_2_O), peak inspiratory pressure (PIP) in cmH_2_O, and respiratory system compliance (in mL/cmH_2_O). These variables included an associated date and time value. Given the complete capture of ventilatory changes inherent in the electronic data warehouse, variables describing ventilatory parameters such as PEEP or V_T_ were assumed to be unchanged until a subsequent value was entered. For variables that remained unchanged for >24 h, we set all values after 24 h to missing. As patients had data recorded with every change or at least every 4 h if no change, this helped eliminate data errors, as data that had no changes at all for 24 h were felt to be erroneous. Further, as we limited our sensitivity analyses to the period of transition, data outside of 24 h after a transition was effectively eliminated. Derived values were then utilized for analysis of all models and tables, with counts of original observations of each variable reported in the results and below each table.

Calculated variables included (1) PaO_2_/FiO_2_ ratio; (2) control versus spontaneous mode categories: “control” included pressure control and volume control, spontaneous included spontaneous and T-piece; and (3) the elastic component of mechanical power. The elastic component of mechanical power has been previously suggested to be a reliable and accurate proxy for the energy applied to the lung during mechanical ventilation that is valid among both paralyzed and spontaneously breathing patients [12]. This is in contrast to the full mechanical power value, which is not valid in spontaneously breathing patients [13]. The elastic component of mechanical power was calculated as:½ × *lung elastance* × (*V_T[in liters]_*^2^) × *respiratory rate*(1)

The first value of any variable per patient was collected at the moment of the start of full support or spontaneous mode, as defined above, after intubation, and the last value collected as the final recorded value prior to extubation. We did not report clinical variables beyond ventilatory settings during the periods defined in each analysis.

### 2.3. Outcomes

We descriptively report ventilatory parameters over time centering around the transition from control modes to spontaneous modes. Outcomes included PaO_2_/FiO_2_ ratio, respiratory rate, V_T_, PEEP, respiratory system compliance, and elastic component of mechanical power. We examined the association between these variables, examining their change over the duration of time since the switch from control mode to spontaneous mode.

### 2.4. Statistical Analysis

We compared outcomes by ventilator mode (control versus spontaneous). Outcomes are descriptively reported as median and interquartile range (IQR), with differences assessed by univariate mixed effects panel regression, clustered by patient. Variables are graphically displayed by ventilator mode over time. For graphs showing duration of time at PEEP, V_T_ and mechanical power (elastic component) as a function of ventilator mode, the duration of time at each value was calculated as median per patient and reported as median plus 95% confidence interval (95% CI) by mode. As our primary goal was to demonstrate the changes within patients based on mode, not between patients, we performed multivariate regression with mixed effects, clustered by patient. This is a type of multilevel regression that allows for analysis of data that is correlated within individual patients [14]. We intentionally did not adjust for or report covariates likely to influence differences between patients, such as age, comorbidities, or etiology of cardiopulmonary failure. We adjusted outcomes for potential confounders within patients, which we had determined a priori based on physiologic relevance and availability. Covariates for adjustment varied based on the outcome for each figure, but were selected from duration of time (in hours) since the switch to spontaneous mode, PEEP, compliance, respiratory rate, and V_T_. Full models are reported in the Appendix A.

As we did not have data on patient sedation level, or other clinical status, we cannot attribute our findings only to the change in mode, but rather we want to highlight that the change in mode may have occurred, and likely did, in response to a change in patient clinical status. As such, the mode may have been a mediator in a causal pathway. Ventilatory changes may have also been associated with changes in clinical status or level of alertness.

### 2.5. Sensitivity Analyses

We performed two sensitivity analyses to assess the durability of our findings. First, as our focus was on the changes that occurred with the switch to spontaneous mode, we performed subset analyses to isolate our analysis on changes that were temporally proximate to the transition of mode; this included isolating from 2 h prior to the switch until 24 h after, or from 12 h prior until 24 h after. This accomplishes a few goals. This process ensured that the numbers of values per patient were balanced, effectively normalized for comparable durations of observation. This temporal restriction enables us to mitigate but not eliminate reverse causality, by focusing on ventilatory findings occurring temporally after the transition to spontaneous mode. As such, the transition occurred temporally first, and the ventilatory settings during spontaneous mode occurred temporally after. Variables were also labeled as occurring during spontaneous mode, additionally enabling identification during the period after the transition. Nevertheless, the findings are associations, and we cannot infer directionality beyond this temporal restriction. The temporal restriction also eliminated variance in duration of data collection per patient, normalizing the imbalance in numbers of observations between patients.

Secondly, as our cohort intentionally included patients across a range of cardiopulmonary failure etiologies requiring ECMO, we additionally performed sensitivity analyses to assess for differences when limited to patients with lung injury, such as acute respiratory distress syndrome. We defined this subset by limiting only to patients who had poor lung compliance (≤30 mL/cmH_2_O) (healthy range 100–400 mL/cmH_2_O) on first ventilatory assessment. From our previous work among patients with ARDS from three clinical trials [15], the median (IQR) lung compliance among patients was 29 mL/cmH_2_O (22,38). In other studies, ventilated patients *primarily without* ARDS (~10% ARDS) had a compliance of 37–111 (IQR) [16]. These studies together suggested to us that initial compliance of ≤30 mL/cmH_2_O was an appropriate cut-off to identify this subgroup.

All statistical analyses were conducted in STATA v.15.1 (College Park, TX, USA). Given the size of the dataset to minimize type I error (false positive), statistical significance was set at the 0.001 level, and all tests were two-tailed.

## 3. Results

Among 483 patients initially, after filtering we had 419 patients for analysis. Among these patients, there were 33,940 original observations of V_T_; 36,882 original observations of respiratory rate; 33,655 original observations of PIP; 10,783 original observations of respiratory system compliance; 36,045 original observations of PEEP; 101,949 *calculated* observations of the elastic component of mechanical power. Appendix A describes variables across the entire cohort, differences in variables between spontaneous modes and control modes, and the statistical significance of the difference for each variable. Briefly, after adjusting for patient clustering, patients on spontaneous mode, versus (vs.) control modes, exhibited greater V_T_, improved compliance, and greater mechanical power (elastic component; range 0 to 1.4).

Results are presented among all patients during the period of transition (Section 3.1), among patients with low compliance (Section 3.2), among patients with low compliance during the period of transition (Section 3.3). Graphical results are reported in Section 3.4, with changes in variables as a function of time in Section 3.5.

### 3.1. Period of Transition

On average, patients underwent a transition from control modes to spontaneous modes 5 (2,8) times. Table 1 describes differences in variables during the transition from control to spontaneous modes (from 12 h prior until 24 h after). Briefly, after adjusting for patient clustering, during spontaneous mode breathing vs. control mode breathing, patients exhibited greater tidal volume (480 mL (375,570.5) vs. 520 mL (411.5,638); *p* < 0.0001), mechanical power (elastic component) (0.08 mL/(cmH_2_O × min) (0.05,0.12) vs. 0.06 mL/(cmH_2_O × min) (0.04,0.09); *p* < 0.0001) (range 0 to 1.4) and respiratory rate (18 (14,22) vs. 21 (16,27); *p* = 0.004). During this time, respiratory system compliance was greater and PEEP was statistically significantly lower, but with minimal clinical difference.

### 3.2. Subset Analysis, Patients with Lung Compliance <30 mL/cmH_2_O

Limited to patients with initial lung compliance <30 mL/cmH_2_O, Appendix A describes differences in variables by control vs. spontaneous modes. Again, during spontaneous mode breathing vs. control mode breathing, patients exhibited greater V_T_ (469 mL (383,581) vs. 375 mL (260,480); *p* < 0.0001), and greater mechanical power (elastic component) (0.08 mL/(cmH_2_O × min) (0.05,0.12) vs. 0.05 mL/(cmH_2_O × min) (0.02,0.09); *p* < 0.0001) (range 0 to 1.4), and lower PEEP (6 cmH_2_O (5,8) vs. 10 cmH_2_O (8,11); *p* < 0.0001).

### 3.3. Subset Analysis, Patients with Lung Compliance <30 mL/cmH_2_O and during the Period of Transition

Limited to patients with initial lung compliance <30 mL/cmH_2_O, and only during the period of transition (12 h prior to 24 h after), Table 2 describes differences in variables by control vs. spontaneous modes. Again, during spontaneous mode breathing vs. control mode breathing, patients exhibited greater V_T_ (471 mL (364,585) vs. 425 mL (320,527); *p* < 0.0001), higher respiratory rate (23 bpm (18,28) vs. 18 bpm (14,23); *p* = 0.003), and greater mechanical power (elastic component) (0.08 mL/(cmH_2_O × min) (0.05,0.12) vs. 0.06 mL/(cmH_2_O × min) (0.03,0.09); *p* < 0.0001) (range 0 to 1.4).

### 3.4. Graphical Analysis

Figure 1 shows the duration of time (median, 95% CI) per patient at each PEEP range. It can be seen that for patients on spontaneous modes, compared to control modes, significantly more time was spent at lower PEEP levels.

Figure 2 shows the duration of time (median, 95% CI) per patient at each tidal volume range during the transition to spontaneous breathing among all patients. Appendix A reports the full multivariate model among all patients, and Appendix A reports it among the subset of patients with low compliance. Patients on control modes spent greater time at higher tidal volumes (300–1000+) compared to control modes.

Figure 3 shows the duration of time (median, 95% CI) per patient at each level of mechanical power (elastic component) range. The main figure shows the significantly greater time at the lowest mechanical power level for patients on control modes, and that above this level, the inset shows that patients on spontaneous mode had roughly double the duration of time at each higher mechanical power level.

### 3.5. Changes in Variables as a Function of Time

To examine the relationship between respiratory rate over time since the transition to spontaneous mode, we graphed the median (95% CI) respiratory rate of patients, after the switch to spontaneous modes. Figure 4 shows the adjusted increase in respiratory rate increases over time since the transition from control mode to spontaneous mode, among the subset of patients with low compliance. Appendix A shows this same adjusted increase among all patients.

To examine the change in PaO_2_/FiO_2_ over time after the change to spontaneous modes, we graphed the median (95% CI) across patients. Figure 5 shows that, among patients who are tachypneic (respiratory rate ≥ 30), as duration of time progresses after the transition from control modes to spontaneous modes, after multivariate adjustment with mixed effects clustered by patient, PaO_2_/FiO_2_ steadily decreases.

This adjusted decrease in PaO_2_/FiO_2_ remained true in subset analysis among patients with low compliance (Appendix A). Decreasing PaO_2_/FiO_2_ with increasing duration of time was only true for patients with tachypnea though. Appendix A shows that among patients without tachypnea, PaO_2_/FiO_2_ did not decrease with time in the overall cohort; among patients with low compliance, it significantly increased (Appendix A). This finding of decreasing PaO_2_/FiO_2_ only among tachypneic patients, and no change or improving PaO_2_/FiO_2_ among patients without tachypnea likely reflects worse lung function and injury in the tachypneic group.

Given our observations on increasing tidal volume and respiratory rate over time, we sought to examine whether the combination of large tidal volumes and increased respiratory rates would be associated with worsened oxygenation, due to development of alveolar injury and edema. Figure 6 shows the adjusted change in PaO_2_/FiO_2_ as a function of the combination of respiratory rate and tidal volume among patients on control modes. It can be seen that the combination of increased tidal volume with increased respiratory rate results in significantly decreased PaO_2_/FiO_2._ This observation is visually apparent at respiratory rates >25, and magnified with the combination with high tidal volume. The relationships between respiratory rate and PaO_2_/FiO_2_ and between tidal volume and PaO_2_/FiO_2_ remained qualitatively unchanged, and the variables were still statistically significantly different among the low compliance subset (Appendix A).

## 4. Discussion

We found that beginning with the transition from control mode to spontaneous modes during mechanical ventilation while on ECMO, patients demonstrated overall increased respiratory rates, increased tidal volumes, and increased elastic component of mechanical power. We further observed that increasing tidal volume and increasing respiratory rate were both temporally associated with decreased PaO_2_/FiO_2_ ratio, even after adjustment. Finally, we found that the combination of increased tidal volume and increased respiratory rate were additive, with a significantly decreased PaO_2_/FiO_2_ among patients with increases in both, especially above respiratory rates of 25.

These findings are the first, to our knowledge, to demonstrate the fixed effect (within patients) of changes in ventilatory parameters while transitioning from control modes to spontaneous modes on ECMO. This transition is ubiquitous among patients weaning from mechanical ventilation, and we found that within this cohort, patients underwent this transition a median of 5 times. The commonality of this transition among this cohort may reflect the ability for ECMO to provide adequate gas exchange despite lower ventilatory settings, and enabling patients to be more alert, both of which may be associated with a transition to spontaneous modes. Our findings are significant in that they suggest that this transition to spontaneous mode may be injurious for many patients, with observed decreases in PaO_2_/FiO_2_ ratio, and increases in tidal volume, respiratory rate, and the elastic component of mechanical power. We should note though that the ARMA study demonstrated worsening PaO_2_ in the group with improved mortality [5], reminding us that we cannot attribute the decreasing PaO_2_/FiO_2_ in this study to worsened lung function or harm based on this finding alone. Additionally, as we did not analyze data on dynamic changes in ECMO support, we cannot exclude the fact that the ECMO circuit may have influenced the change in PaO_2_/FiO_2_.

Previous studies have demonstrated that spontaneous breathing during mechanical ventilation for ARDS induces a pendelluft phenomenon within the lung [17], in which volumes of gas transition from one region to another without passing through the endotracheal tube, potentially resulting in occult regional over and under distension. Further, spontaneous breathing necessitates active muscle contraction, distorting the relationship between pressures measured at the endotracheal tube and those experienced by regional areas of the lung [18,19]. Thus, the ventilatory measurements, such as PEEP, may be inaccurate in spontaneous breathing patients. To address these limitations, we measured the elastic component of mechanical power, which has been suggested to be a superior method to quantify the energy applied to the lung in patients who are spontaneously breathing [12].

Our findings in this study thus build upon our previous work examining the association of mechanical power and outcomes among patients with ARDS [15]. It has been previously suggested that mechanical power, which encompasses additional ventilatory parameters such as rate, is able to capture repetitive force in a way that static measures are not [15,20,21]. In our previous work, we demonstrated that mechanical power added additional mortality prediction over other measures of force [15]. In this study, we showed that among spontaneously breathing patients, the elastic component of mechanical power was significantly higher among spontaneously breathing patients. If the elastic component of mechanical power is associated with lung injury or patient mortality, then this finding has significant implications. Our study also builds upon important previous knowledge that a period of cyclic overinflation can cause pulmonary edema and lung injury [22,23]. Our study shows that among patients with ECMO and mechanical ventilation, the combination of overinflation and rapid respiratory rate was temporally associated with significantly decreased PaO_2_/FiO_2_ ratio. Given our findings are retrospective and show associations only, prospective observational and clinical trials are needed to confirm or refute the validity of our findings.

### 4.1. Limitations

Our study has a number important limitations. First, our study is largely descriptive of ventilatory observations, and should not suggest causality. Specifically, the factors driving changes in ventilatory parameters were not assessed in this study, and could include changes in metabolic demand (development of sepsis, hemorrhage), changes in sedation, or fluid administration, or worsening of lung injury. These changes may have contributed to the observed changes in ventilatory parameters. We attempted to adjust for relevant confounders for many of our observations, but we recognize that we did not examine all clinical drivers of changes in ventilation or pulmonary function. For instance, as partial pressure of carbon dioxide (PaCO_2_), a major driver of minute ventilation, was clinically modified though protocoled regular adjustments to sweep gas through the ECMO circuit, we did not adjust for degree of ventilatory failure/PaCO_2_, which remains an unmeasured covariate. Concomitantly, another important limitation is that as PaO_2_ could certainly be influenced by ECMO flow rates, we cannot attribute changes in blood oxygenation only to ventilatory changes, as it may have been influenced by ECMO flow or hemoglobin level, which was not available to us.

Secondly, we did not examine global outcomes such as length of stay, mortality or duration of mechanical ventilation. Further studies should examine whether the observed worsening of ventilatory parameters had an association with these important clinical outcomes. It is possible that these changes were not associated with worsened clinical outcome, and were simply reflective of pulmonary mechanical changes during weaning in the setting of acute lung injury.

Thirdly, we appreciate that in this single center analysis the ventilator parameters may be systematically influenced by institutional behavior. We cannot overcome this, but we tried to mitigate the effect by analyzing ~10 years of data, containing over 400 patients managed with ECMO for a variety of indications, in the hope that the temporal span and diversity would surmount patterned behavior in providers. We believe that the consistency and magnitude of our findings suggests some truth to the patterns beyond the institution, but our data requires external validation to confirm or refute our findings.

Finally, our analysis is among a heterogenous clinical group of patients with cardiopulmonary failure, including both ARDS, and patients with cardiogenic shock. The observation of significance within this heterogeneous clinical group suggest to us the durability of our findings, which we additionally confirmed by performing subset analysis among patients with low compliance, as would be found in patients with ARDS. Nevertheless, we recognize that this is imperfect method and further studies should prospectively collect data with consideration of these limitations.

### 4.2. Conclusions

In this analysis of 419 patients on mechanical ventilation and ECMO, we demonstrated that there were observable and statistically significant changes in ventilatory settings and blood gases during the transition from full support (control) modes to spontaneous modes. We further demonstrated time dependent changes in ventilator parameters around this period of transition. These changes in ventilator parameters warrant prospective study, as they may be associated with worsened lung injury.

## Figures and Tables

**Figure 1 jcm-10-01001-f001:**
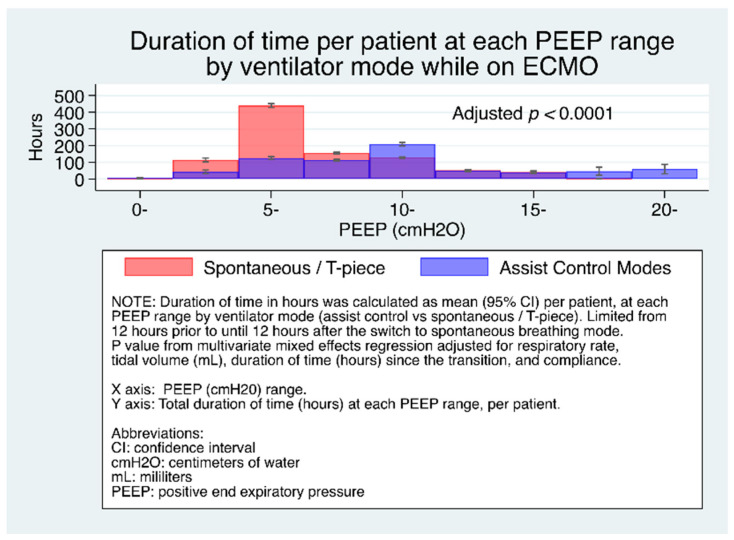
Duration of time (median, 95% CI) per patient at each PEEP range by ventilator mode.

**Figure 2 jcm-10-01001-f002:**
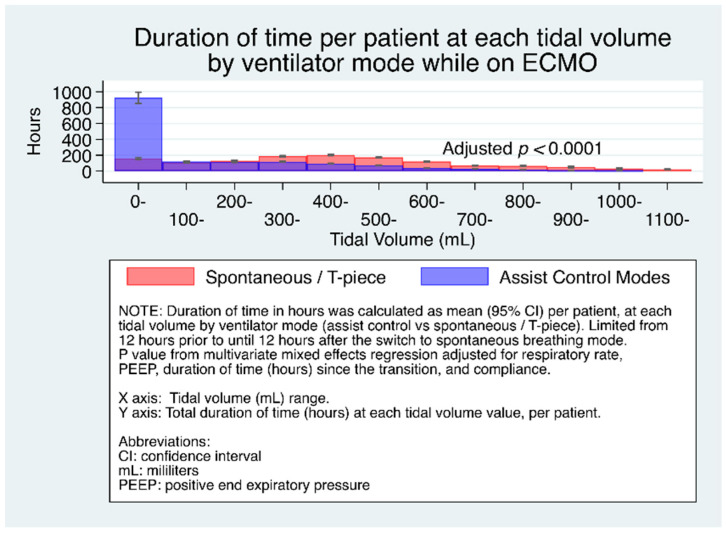
Duration of time (median, 95% CI) per patient at each tidal volume range by ventilator mode.

**Figure 3 jcm-10-01001-f003:**
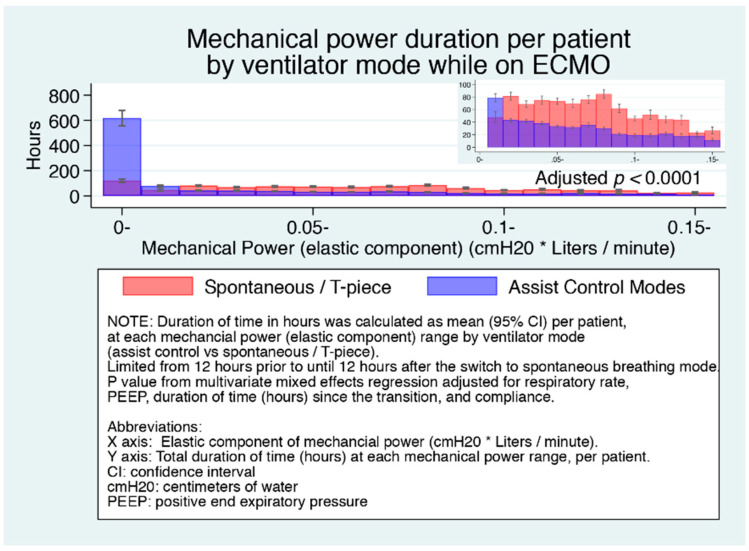
Duration of time (median, 95% CI) per patient at each mechanical power (elastic component) range by ventilator mode.

**Figure 4 jcm-10-01001-f004:**
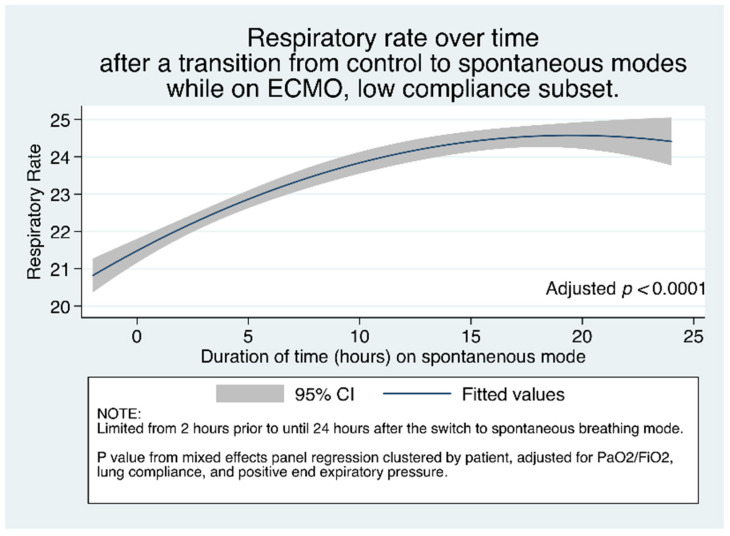
Respiratory rate over time after a transition from control to spontaneous modes while on ECMO, among patients with low compliance (<30 mL/cmH_2_O). *p* value reflects multivariate mixed effect regression, clustered by patient.

**Figure 5 jcm-10-01001-f005:**
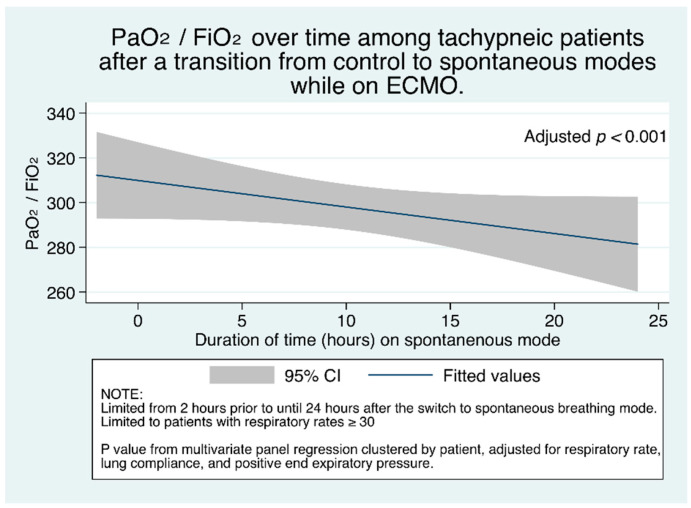
PaO_2_/FiO_2_ ratio (95% CI) over time among tachypneic patients (respiratory rate ≥ 30).

**Figure 6 jcm-10-01001-f006:**
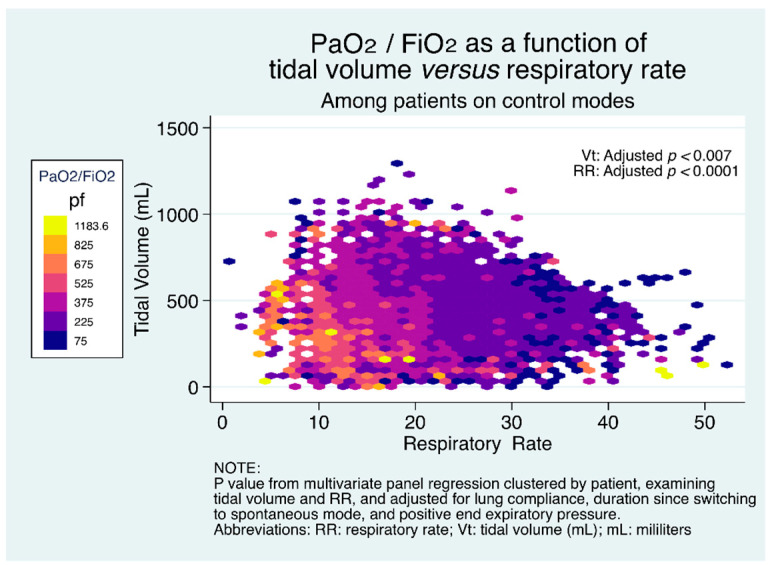
PaO_2_/FiO_2_ as a function of the combination of respiratory rate and tidal volume among patients on control mode.

**Table 1 jcm-10-01001-t001:** Ventilator parameters during the transition from control to support modes.

Variable ^1^	All	Control Modes	Spontaneous Modes	*p-*Value ^2^
Tidal volume (mL)	504 (400,611)	480 (375,570.5)	520 (411.5,638)	<0.0001
Respiratory rate (breaths per minute)	20 (16,25)	18 (14,22)	21 (16,27)	0.004
Peak inspiratory pressure (cmH_2_O)	30 (20,40)	32 (22,40)	28 (19,40)	<0.0001
Respiratory system compliance (mL/cmH_2_O)	34.1 (25.1,43.5)	33 (24.1,42.5)	34.6 (25.8,43.9)	<0.0001
Positive end expiratory pressure (cmH_2_O)	8 (5,10)	8 (5.8,10)	7.8 (5,9.9)	<0.0001
Elastic component of mechanical power (mL/(cmH_2_O × min)	0.07 (0.05,0.11)	0.06 (0.04,0.09)	0.08 (0.05,0.12)	<0.0001

^1^ median, interquartile range (IQR); Abbreviations: cmH_2_O: centimeter of water; mL: milliliter; min: minute; Number of original observations: Tidal volume: 8106; Respiratory rate: 8826; Peak inspiratory pressure: 8123; Respiratory system compliance: 1703; Positive end expiratory pressure: 9121. Elastic component of mechanical power: 21,732 calculated observations. ^2^
*p* value from univariate mixed effects panel regression model of one value per hour, clustered by patient.

**Table 2 jcm-10-01001-t002:** Subset analysis of ventilator parameters during the transition from control to support modes, among patients with lung compliance <30 cmH_2_O.

Variable ^1^	All	Control Modes	Spontaneous Modes	*p*-Value ^2^
Tidal volume (mL)	453 (347,567)	425 (320,527)	471 (364,585)	<0.0001
Respiratory rate (breaths per minute)	21 (16,26)	18 (14,23)	23 (18,28)	0.003
Peak inspiratory pressure (cmH_2_O)	30 (21,40)	33 (23,40)	29 (20,40)	<0.0001
Respiratory system compliance (mL/cmH_2_O)	29 (21.6,39.1)	27.8 (20,36.6)	29.7 (22.3,40)	0.005
Positive end expiratory pressure (cmH_2_O)	8 (5.1,10)	8.4 (6.8,10)	8 (5,10)	<0.0001
Elastic component of mechanical power (mL/(cmH_2_O × min)	0.07 (0.04,0.11)	0.06 (0.03,0.09)	0.08 (0.05,0.12)	<0.0001

^1^ median, interquartile range (IQR); Abbreviations: cmH_2_O: centimeter of water; mL: milliliter; min: minute; Number of original observations: Tidal volume: 16,514; Respiratory rate: 18,025; Peak inspiratory pressure: 16,454; Respiratory system compliance: 5496; Positive end expiratory pressure: 17,566. Elastic component of mechanical power: 51,694 calculated observations. ^2^
*p*-value from univariate mixed effects panel regression model of one value per hour, clustered by patient.

## Data Availability

Data code is available from the corresponding author upon reason- able request.

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
