# Peer review of "On the Transition from Control Modes to Spontaneous Modes during ECMO"

_jcm, 2021, doi:10.3390/jcm10051001_

Round 1

Reviewer 1 Report

I would like to commend the authors on making the changes that they did.  Decreasing the sample size and years of study significantly addressed my concerns with their study design and improved the validity of the study.  I have no further recommendations or concerns to make.  Thank you for making this contribution to the medical literature and knowledge.

Author Response

REVIEWER 1:

I would like to commend the authors on making the changes that they did.  Decreasing the sample size and years of study significantly addressed my concerns with their study design and improved the validity of the study.  I have no further recommendations or concerns to make.  Thank you for making this contribution to the medical literature and knowledge.

RESPONSE: Thank you sincerely.

Reviewer 2 Report

I think the revisions have strengthened the manuscript and the authors did a good job of addressing the comments. Thank you for letting me review the manuscript.

Author Response

REVIEWER 2:

I think the revisions have strengthened the manuscript and the authors did a good job of addressing the comments. Thank you for letting me review the manuscript.

RESPONSE: Thank you sincerely.

This manuscript is a resubmission of an earlier submission. The following is a list of the peer review reports and author responses from that submission.

Round 1

Reviewer 1 Report

The subject and content of this study is very needed in the field of critical illness and ECMO.  While ECMO has become more widespread, the degree of scientific knowledge to guide evidence-based practice.

Given that ECMO essentially replace or support cardiopulmonary physiology during recovery, the authors sought to determine if the transition from controlled mechanical ventilation to spontaneous ventilation was safe in patients requiring ECMO.  This is not insignificant as the the risk of VILI in these patients is potential cause of morbidity and failure to wean from ECLS.

The authors gathered their data and their information from a vast database of patients and mechanical ventilation parameters to which they have access.  This is definitely commendable.

I do have significant concerns regarding the method used for the study.  The patient population was not a homogenous population.  1) The patient encompassed a study period of 23 years.  During that time there was significant changes to the standards of practice in mechanical ventilation that would prevent using the population as whole to draw a conclusion.  2) The use of ECMO can bet for either cardiac and/or pulmonary failure, both with very different pathophysiologies.  The study population included patients on mechanical circulatory support as well VA and VV ECMO.  The authors tried to accommodate for this with their subgroup analysis but the primary analysis was done with the population at large.  3) Given that the study population was from a 23 year period, there would be concern if there was a consistency in ECLS practice during this time period as well as other standards of care that would impact these patients.

The authors report that the transition to spontaneous ventilation could potentially result in harm with a decrease in P/F ratio for patients.  The authors should be cautioned in drawing this conclusions.  1) Previous studies have shown that a decrease in P/F ratio is simply that and not necessarily connected with patient outcomes.  2) Given the heterogeneity of the population, I would hesitate in making conclusion on the PaO2/FiO2 ratio.  This would require that only mechanical ventilation be changed and ECLS support not be adjusted with every mode change in mechanical ventilation.  If the authors did this is, it should be stated.

Overall, I commend the authors for attempting to address this issue of the most safe method to mechanically ventilate patients while on ECMO but would hesitate in drawing significant conclusions from this study protocol and resulting data. 

Reviewer 2 Report

Summary:
The authors investigated in a cohort of 419 patients undergoing treatment with ECMO the effect of a transition from controlled to spontaneous ventilation mode on a set of outcome variables, including PaO2/FiO2 ratio, Vt, respiratory rate, PEEP, and the elastic component of mechanical power. The study is well written and has been thoroughly conducted. The manuscript was pleasant to read, and the findings are interesting for clinicians and researchers. However, the main concerns are that the exposure and the causal framework are not well defined. Potential reverse causality makes the interpretation of the clinical relevance of the findings very difficult. Furthermore, there might be a problem with the statistical analysis approach.

Major:
1. The paper would benefit from a clear description of the underlying causal concept or framework. It is challenging to determine the relevance of the described changes and patterns in ventilatory variables. Many of the reported effects could be subject to reverse causality. The shift from a controlled to a spontaneous ventilation mode is most often the response to a patient state change. For example, the reported decrease in PaO2/FiO2 ratio and the respiratory rate changes after transition might be related to weaning sedation and blood or sweep gas flow changes. It remains unclear whether the authors frame the transition from controlled to spontaneous ventilation as an intervention/exposure or as an intermediate variable on the pathway to the outcomes. The intervention or exposure being studied in the present work is not well defined. Thereby the reader cannot fully appreciate the clinical relevance of the observed patterns for the outcomes measured.

2. Some of the analyses were adjusted for confounders. However, a more detailed description of how these confounders were selected is needed. Instead of "data-driven" selection, the use and definition of a causal framework would be helpful. As an example of why this is relevant: the authors describe the time-dependent change in respiratory rate after the transition from controlled to spontaneous mode (Figure 4). The authors adjusted the analysis for the degree of hypoxemic respiratory failure and lung compliance. However, no adjustments have been made for the degree of ventilatory failure. What was the reason to only adjust for one part of the respiratory failure?

3. The authors compared medians of outcome variables stratified by the level of the exposure variable. Furthermore, they describe that student's t-tests have been used for hypothesis testing. This approach could be problematic as pre- (controlled mode), and post- (spontaneous mode) measurements are dependent. Also, individual patients will contribute varying numbers of observations to calculate the median or mean, depending on the time spent in the respective mode. Maybe weighting or adjusting of the median or mean would be necessary to obtain reliable estimates?

4. During the first time reading the paper, I was confused why the authors did not use mixed-effects models with random slope and intercept (assuming no informative missingness; otherwise, methods like joint models would be needed). This would have facilitated the estimation of crude and adjusted average changes in the outcome variables based on a change in exposure level. The panel models used in this article are often applied for the analysis of economic data. However, medical researchers are not very familiar with these types of models. In the Methods section, the step from describing medians/student's t-test to panel models is steep and a bit surprising. More explanation is needed on why this methodology has been chosen (e.g., repeated measurements and multiple transitioning episodes, unbalanced panel sets?). However, if the reason leading to using panel models was that the panel sets were not balanced, why did the authors also use medians and students' t-tests for comparisons? This needs a better introduction/explanation. Also, the authors should report their model results in a supplementary table.

5. Please consider defining the start and duration of follow-up in the Methods section. Also, the granularity of the measurements needs to be reported. Are the number of measurements the same for all patients? Is there potential informative missingness due to prognosis (missingness due to extubation; missingness due to differential measurements, e.g., PaO2/FiO2 ratio might be more frequently measured in more critically ill patients)? How did the authors deal with this?

6. Please clarify the text in the Methods section, page 2, line 88. It remains unclear for the reader why variables that were unchanged for at least 24 hours were reassigned as missing values. Does this introduce bias?

7. The PaO2/FiO2 ratio could be substantially influenced by FdO2 of the ECMO and blood flow. ECMO settings might have been changed during transitioning to spontaneous breathing. How did the authors address this problem?

Minor:
1. I am sure that this due to a typo: the authors reported that 101,949 observations were available for the elastic component of mechanical power. However, only 33,940 observations have been recorded for tidal volume.

2. Maybe, the authors could consider to report the changes in tidal volumes in mL per predicted body weight. Usually, tidal volumes are set based on predicted body weight. It is difficult to interpret the clinical relevance of a change in absolute tidal volumes between exposure levels.

3. Please clarify: was it dynamic or static elastance that has been measured?

4. At which time point did the first ventilatory assessment took place (the measurement of compliance for subsequent stratification)? This should be defined and stated in the methods, like for the start of follow-up.

Reviewer 3 Report

Thank you for the giving me the opportunity to review the article "on the transition from control modes to spontaneous mode during ECMO"

In this study investigators have reviewed data from 419 mechanically ventilated patients on ECMO over 20 years in a single center. They examined more than 30,000 ventilator variables on these patients. Investigators objective was to determine if there was any difference between ventilator parameters during spontaneous modes compared to the control modes and if these differences were associated with changes in the oxygenation/ventilation.

If I am understanding it correctly the hypothesis is that the unregulated breeding during spontaneous breathing may result in worsening of lung injury.

(1) I have a hard time understanding what is meant by "transition from control mode to spontaneous mode". Patients on ventilator with or without ECMO may be on different ventilator settings. The only true spontaneous mode would be when the patient is only on CPAP pressure support mode of ventilation. Otherwise the patient can be on assist control SIMV, APRV etc..

So either the investigators are separately analyzing CPAP/pressure support mode of ventilation with all controlled motor ventilation or they are separately assessing the ventilator breaths and spontaneous breaths.

(2) investigators have done a commendable job in cleaning the data and analysis. Missing data and sensitivity analysis was also performed

(3) investigators found that the patient and spontaneous modes had higher tidal volumes (513 ML versus 428 ML), higher respiratory rate (22 breaths per minute versus 18 breaths per minute, lower peep 5.4 versus 10 and improved compliance (35 mL/cmH2O versus 28 mL/cm water)

I would assume that the patient would be transition to a spontaneous mode of ventilation when he is "recovering" so finding a higher tidal volume and improved compliance is not surprising. Similarly peep is set by the physician and it is also expected that the patient who is improving would have a lower peep.

(4) investigators have also not given any ECMO parameters. On ECMO PF ratio would very much depend on what the oxygenation from ECMO circuit is.

Is it possible that when the patient is improving, patient is being weaned from the ECMO circuit also thereby patient is getting less support from ECMO which would mean lower PF ratio.

(5) the only outcome difference investigators can show is a difference in the PF ratio. It would be easy enough to show or compared the patient variables like duration of ventilation which may provide more meaning to subtle changes in the ventilator parameters.